# Enhanced Hybrid Nanogenerator Based on PVDF-HFP and PAN/BTO Coaxially Structured Electrospun Nanofiber

**DOI:** 10.3390/mi15091171

**Published:** 2024-09-21

**Authors:** Jin-Uk Yoo, Dong-Hyun Kim, Eun-Su Jung, Tae-Min Choi, Hwa-Rim Lee, Sung-Gyu Pyo

**Affiliations:** School of Integrative Engineering, Chung-Ang University, 84, Heukseok-ro, Dongjak-gu, Seoul 06974, Republic of Korea; wlsdnr5771@naver.com (J.-U.Y.);

**Keywords:** electrospinning, coaxial structure, energy harvesting, piezoelectric effect, triboelectric effect

## Abstract

Nanogenerators have garnered significant interest as environmentally friendly and potential energy-harvesting systems. Nanogenerators can be broadly classified into piezo-, tribo-, and hybrid nanogenerators. The hybrid nanogenerator used in this experiment is a nanogenerator that uses both piezo and tribo effects. These hybrid nanogenerators have the potential to be used in wearable electronics, health monitoring, IoT devices, and more. In addition, the versatility of the material application in electrospinning makes it an ideal complement to hybrid nanogenerators. However, despite their potential, several experimental variables, biocompatibility, and harvesting efficiency require improvement in the research field. In particular, maximizing the output voltage of the fibers is a significant challenge. Based on this premise, this study aims to characterize hybrid nanogenerators (HNGs) with varied structures and material combinations, with a focus on identifying HNGs that exhibit superior piezoelectric- and triboelectric-induced voltage. In this study, several HNGs based on coaxial structures were fabricated via electrospinning. PVDF-HFP and PAN, known for their remarkable electrospinning properties, were used as the primary materials. Six combinations of these two materials were fabricated and categorized into homo and hetero groups based on their composition. The output voltage of the hetero group surpassed that of the homo group, primarily because of the triboelectric-induced voltage. Specifically, the overall output voltage of the hetero group was higher. In addition, the combination group with the most favorable voltage characteristics combined PVDF-HFP@PAN(BTO) and PAN hollow, boasting an output voltage of approximately 3.5 V.

## 1. Introduction

In modern society, the development of science and technology has led to the commercialization of various efficient energy sources. However, challenges persist with the electrochemical inorganic battery system, primarily due to the use of heavy metals such as nickel and cobalt as well as toxic electrolytes, resulting in environmental pollution issues [1,2]. Therefore, there is growing interest in environmentally friendly energy-harvesting systems as potential replacements, garnering significant attention [3,4,5]. Energy-harvesting systems are technologies that convert wasted resources into electrical energy, offering an essential alternative to traditional energy supply methods. These systems facilitate the operation of various devices by converting inefficiently wasted solar, wind, thermal energy, and mechanical motion into usable electrical energy. In addition, they enable the powering of devices using eco-friendly and sustainable energy sources.

Furthermore, as technology advanced to enhance the portability and efficiency of devices, environmental concerns and challenges associated with the miniaturization of electronic devices were encountered. The development and commercialization of wearable devices have underscored the need for flexible materials. Therefore, electrospun hybrid nanogenerators have emerged as a promising solution to address these issues and are suitable energy-harvesting systems for wearable electronics, health monitoring, IoT devices, etc. In addition, electrospinning has the advantages of being environmentally friendly, facilitating miniaturization, and providing flexibility [2,6,7,8].

Electrospinning involves the production of solid-phase fibers by applying a high voltage to a polymer solution. This method enables the production of low-cost nano-scale fibers and offers the advantages of material versatility selection and processing simplicity. Consequently, electrospinning has proven valuable in the production of HNGs [9,10,11]. However, the performance of HNGs studied in recent years is insufficient for commercial use in various applications. Therefore, this work aims to improve the harvesting ability by producing fibers with a core–shell or hollow structure via electrospinning, based on the fact that nanofibers with a coaxial structure exhibit superior bending deformation properties compared to single fibers [2,12,13]. For material selection, fibers were fabricated using PVDF-HFP and PAN, both of which are renowned for their outstanding piezoelectric characteristics [14,15,16,17].

In conclusion, an energy-harvesting system was successfully fabricated by combining piezoelectric and triboelectric effects using the methods and materials mentioned above. Piezoelectric nanogenerators (PENGs) [18,19] and triboelectric nanogenerators (TENGs) [20,21,22] offer distinct advantages, including independence from location, weather, environment, and time constraints. In addition, they benefit from a diverse range of materials and simple fabrication methods. Therefore, ongoing research is dedicated to exploring the application of hybrid nanogenerators as energy sources for wearable devices [23,24,25].

## 2. Results and Discussion

In general, PVDF-HFP and PAN have high piezoelectric effect and mechanical strength. PVDF-HFP has a lower piezoelectric effect compared to PAN but has a much higher mechanical strength. Therefore, to take advantage of the advantages of both materials, we studied fibers with a core–shell structure. In addition, to realize a hybrid nanogenerator, it is necessary to effectively show the triboelectric effect between the two meshes. Therefore, in this study, coaxial electrospun nanofibers with different PAN@PVDF-HFP and PVDF-HFP@PAN structures were fabricated by changing the core and shell structures.

Figure 1a shows an SEM image of PAN(BTO)@PVDF-HFP. Given that the shell is composed of PVDF-HFP, the BTO doped on the PAN is not visible and has a smooth surface. However, there is a significant difference in the fiber diameter, resulting in poor fiber diameter uniformity. In contrast, PVDF-HFP@PAN (BTO) (Figure 1b) and PAN hollow fibers (Figure 1c) exhibit highly uniform fiber diameters. In addition, both fibers present a granular surface with BTO doping because the shell consists of PAN (BTO). Figure 2 illustrates a cross-section of a PAN hollow fiber with a well-formed hollow structure. Similar to Figure 1b,c, Figure 2 shows that the BTOs are well dispersed in the PAN fibers.

The crystal structure was confirmed by X-ray diffraction. Because all three types of fibers were composed of PAN with BaTiO_3_ as a dopant, the diffraction pattern of $BaTiO_{3}$ can be observed in Figure 3. The diffraction peaks of BaTiO_3_ nanoparticles appeared sharp and distinct at 22, 31, 39, 43, 54, and 65°, corresponding to the (100), (110), (111), (002), (211), and (220) planes, respectively [26]. These peaks closely align with the tetragonal phase of $BaTiO_{3}$, indicating successful doping. In addition, significant diffraction peaks of PVDF-HFP in PAN@HFP with PVDF-HFP as the shell were observed compared to HFP@PAN and PAN hollow (Figure 3b). The peaks at 18.6° and 20.1° correspond the α-phase and β-phase of PVDF-HFP, respectively, exhibiting a broader full width at half maximum (FWHM) than the peaks of BaTiO_3_. The FWHM of a phase indicates its crystallinity. The broader the pick, the lower the crystallinity [27]. It can be expected that the higher the crystallinity of the β phase of PVDF-HFP, the greater the piezoelectric effect. The overall peak size exhibited a decrease after the cyclic tensile tests, indicating a reduction in crystallinity induced by fatigue [28].

Initially, the output voltage of the primary fiber was compared, serving as the basis for the hybrid nanogenerator (Figure 4). The output voltage corresponds to the peak-to-peak-alternating voltage induced by the cyclic tensile forces. The output voltages of PVDF-HFP, PAN, BTO-doped PAN nanofibers fabricated via single electrospinning and coaxial electrospun nanofibers are different. In general, the output voltage of coaxially structured fibers is better than that of nanofibers fabricated by single electrospinning. The increased surface area is the main reason for this. BTO was doped at 15 wt% relative to PAN; higher doping amounts result in brittle fibers or difficulty in fiber formation (Appendix A). In addition, the output voltage of the PAN fibers at 15 wt% was higher than other doping contents (Appendix A). Consequently, the Fiber A (PAN(BTO)@PVDF-HFP specimen) exhibited a voltage of approximately 400 mV, while the Fiber C (PAN hollow specimen) exhibited a voltage range of 474 to 515 mV. However, Fiber B (PVDF-HFP@PAN(BTO specimen) exhibited a lower voltage level, decreasing from 275 mV to 209 mV. This voltage decrease indicates that the durability of the Fiber B specimen may be insufficient. The lower output voltage is due to the difference in mechanical strength between PVDF-HFP fibers and PAN/BTO fibers (Appendix A). The BTO-doped PAN with relatively low mechanical strength is prone to mechanical deformation due to continuous stretching. Fiber B (PVDF-HFP@PAN(BTO)), with a high relative content of BTO-doped PAN, will continuously decrease the output voltage due to a large amount of PAN(BTO) deformation. The mechanical strength of these fibers are shown in Appendix A.

Finally, the output voltage of the hybrid mesh, based on the primary fiber, was measured and analyzed (Figure 5, Table 1).

Voltage Magnification $\left(=\frac{Voltage\;of\;hybrid\;mesch}{sum\;of\;two\;primary\;fiber^{\prime }smedian\;voltage}\right)$ shows that the increase is more than 1.109 to 4.805 times, except for the combination of Fiber B. This indicates that the triboelectric effect was effectively manifested in most of the combinations.

The triboelectric effect between identical polymers arises from differences in surface depth and composition. In contrast, the triboelectric effect between different polymers exhibits superior electron and material transfer due to the distinct properties of the materials involved, resulting in higher triboelectric-induced voltage [29]. Consequently, the voltage magnification for homo groups comprising identical fibers was less than 1.4 times the output voltage of the base fiber, whereas in hetero groups combining different fibers, the magnification exceeded two times. In conclusion, the combination group incorporating fibers B and C exhibited the highest output voltage characteristics, with an output voltage ranging from 3.431 to 3.541 V.

## 3. Materials and Methods

### 3.1. Materials (Solution Preparation)

A 20 wt% solution of PVDF-HFP was prepared using the following solvent system conditions: PVDF-HFP pellets with a weight-average molecular weight (M_w_) of approximately 400,000 and a number-average molecular weight M_n_ of approximately 130,000 (Sigma Aldrich, Seoul, Republic of Korea) were stirred using a magnetic stirrer (WISD) in a mixture of DMF (Guaranteed Reagent, DAEJUNG, Goryeong, Republic of Korea) and acetone (Guaranteed Reagent, DAEJUNG) in a 5:5 volume ratio at 80 °C for 24 h.

A PAN solution with a molecular weight of approximately 150,000 (Sigma Aldrich) was prepared at a concentration of 10 wt% using DMF as the solvent. In addition, barium titanate (BaTiO_3_) nanopowder, with a particle size of 50 nm (Sigma Aldrich), was used as a dopant to enhance the piezoelectric effect [30,31], with a concentration of 15 wt% compared with the PAN solute. To prepare the PAN (BTO) solution, the BTO dopant was sonicated in an ultrasonic cleaner (DAIHAN Scientific, Wonju, Republic of Korea) to disperse it evenly and then cooled to room temperature after sonication. Subsequently, the sample was stirred on a hotplate. Before this experiment, BTO was doped into the PAN solution according to the weight percent concentration to enhance the piezoelectric effect, which is shown in Appendix A. The experimental results showed that the highest output voltage and fracture stress were measured for PAN fiber doped with 15 wt% compared to PAN. The PAN fiber doped with more than 15 wt% BTO could not form a uniform fiber, and the fiber became very brittle.

### 3.2. Coaxial Electrospinning Process

The objective was to fabricate core–shell structures by combining PVDF-HFP and PAN(BTO) solutions under the conditions mentioned above. From numerous combinations of structures, the focus was placed on three specific structures: PVDF-HFP@PANBTO, PANBTO@PVDF-HFP, and PAN hollow structures, which were expected to exhibit excellent piezoelectric effects.

For the metal nozzle, a coaxial nozzle (nozzle adapter, NanoNC, Seoul, Republic of Korea) was used to produce core–shell and hollow-structured fibers. In addition, a syringe pump (Fusion 100-X Precision Dosing Two-Channel Syringe Pump, Chemyx, Stafford, TX, USA) maintained the core flow rate at 0.5 mL/h and the shell flow rate at 2 mL/h during 2 h of electrospinning. To enhance the piezoelectric effect by increasing the β phase ratio of PVDF-HFP [14,15,32], the fibers were aligned at 600 rpm using a drum-type collector (NNC-DC90H, NanoNC) [29]. The two types of core–shell fibers consisting of PVDF-HFP and PAN(BTO) were maintained at a tip distance of 13.5 cm, with an applied voltage of 19.8 kV from the power supply. If a higher voltage than the critical voltage is applied, more β phase is formed, but the fibers may break or bead during the stretching process. Therefore, PVDF-based fibers with high β phase can be fabricated by applying a high voltage and setting an appropriate distance between the nozzle and tip [33]. For the PANBTO hollow fiber, heavy mineral oil (Sigma Aldrich) was initially injected into the core to produce a hollow structure before electrospinning. During electrospinning, the tip distance was set at 12 cm, with an applied voltage of approximately 22.8 kV. However, during the deposition of the PANBTO hollow fiber onto the rotating drum, the electric field between the fiber and the nozzle became unstable. Therefore, the fabrication process involved gradually reducing the input voltage over time to address the instability issue. Subsequently, the deposited coaxial fiber mesh was immersed overnight in octane (Extra Pure, SAMCHUN, Seoul, Republic of Korea) and dried to facilitate conversion into PANBTO hollow fibers with the core fully removed. Furthermore, throughout the experiment, the tip distance between the nozzle and the drum was maintained, while the metal substrate behind the drum was grounded. An overview of the experimental setup is shown in Figure 6. During the electrospinning process, various parameters (voltage, distance tip to collector, temperature, humidity, etc.) have various effects on the characteristics of the fiber, such as structure, morphology, and diameter. Therefore, the parameters should be adjusted according to the experimental environment.

### 3.3. Characterization Techniques (Cyclic Tensile Tests)

A material-testing machine (Ametek Inc., America Co., Berwyn, PA, USA) was used to measure the piezoelectric voltage generated from the tensile strength testing, as depicted in Figure 7. All measurements were performed using a 3 min cyclic test with a 4 mm extension.

The output voltage of the primary fiber (PVDF-HFP@PANBTO, PANBTO@PVDF-HFP, PAN hollow) was derived from the peak-to-peak value of the alternating voltage measured by grounding both ends, as illustrated in Figure 4.

For the hybrid nanogenerators, six combinations were created by combining the three types of primary fibers selected earlier. During testing, grounding was applied at the end of each fiber. The alternating voltage derived from each of the six combinations was measured and compared, as shown in Figure 5. As the extension progresses, contact was repeatedly made by the electric field between the meshes, resulting in the appearance of triboelectric-induced voltage [34,35]. The triboelectric effect between fibers occurs as the fibers repeatedly come into contact and separate, exchanging electrons, which forms a charge build-up and potential difference. This effect is strongly influenced by the electrophilicity of the fiber material, surface condition, mechanical movement, and environmental conditions [2,22,25]. Consequently, the presence or absence of the triboelectric effect was detected by comparing the voltage of the primary fiber measured earlier.

## 4. Conclusions

We successfully synthesized core–shell and hollow-structured nanofibers with PVDF and PAN(BTO) through a simple electrospinning process, confirming the results with SEM images and XRD diffraction peaks. The output voltage of the hybrid nanogenerators, based on triboelectric and piezoelectric effects, was measured using cyclic tensile tests, with XRD confirming the crystallinity before and after the tests. The analysis of the output voltage based on the measured values revealed significant differences between the homo and hetero groups. The hetero group exhibited heightened triboelectric effects because of the carrying-mesh composition. Notably, PVDF-HFP@PAN(BTO) combined with PAN hollow structures within the hetero group displayed superior voltage characteristics among the six samples tested. In addition, XRD analysis of crystallinity before and after the cyclic test indicated an overall reduction in the diffraction peaks due to mechanical cyclic strength.

Piezoelectric nanogenerators (PENGs) and triboelectric nanogenerators (TENGs) are garnering significant interest as eco-friendly energy-harvesting systems. In addition, electrospinning is gaining traction as an effective method to enhance the voltage generation of hybrid nanogenerators that combine PENGs and TENGs due to its freedom of material selection.

In electrospinning experiments, various parameters, including solution condition, distance to tip, voltage, flow rate, temperature, and humidity, play crucial roles in determining the performance of the resulting fiber and mesh. In particular, it is essential to optimize these materials to determine the structure and material of the fiber because they influence properties such as electrical affinity, transfer media, mechanical strength, and dielectric features. Therefore, achieving high-output voltage and voltage stability with specific hybrid nanogenerators based on coaxially structured electrospun nanofibers indicates the potential for developing efficient energy-harvesting systems. Hybrid nanogenerators are expected to become an important energy-harvesting technology for wearable devices, as well as for various applications such as healthcare, environmental monitoring, and smart cities, and continued research into new materials, structures, and electrospinning processes, such as the coaxial structure and BTO addition in this study, is necessary to develop these technologies into more efficient and reliable energy sources.

## Figures and Tables

**Figure 1 micromachines-15-01171-f001:**
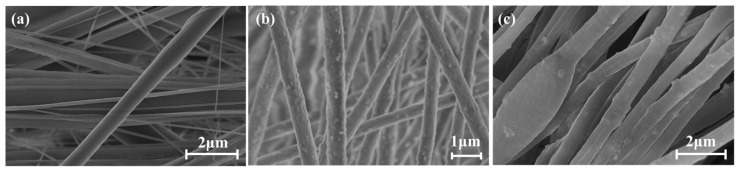
SEM images of (**a**) PAN(BTO)@PVDF-HFP, (**b**) PVDF-HFP@PAN(BTO), (**c**) SEM image of the PAN(BTO) hollow fibers.

**Figure 2 micromachines-15-01171-f002:**
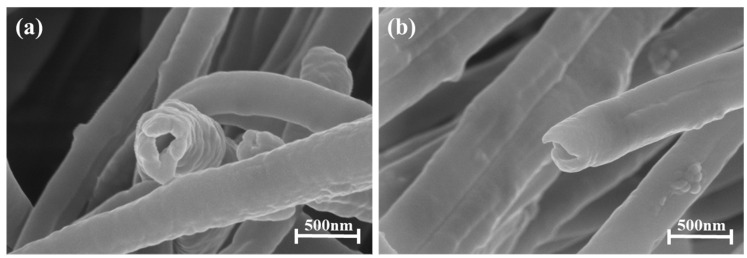
SEM image of the cross-section of the (**a**), (**b**) PAN(BTO) hollow fibers.

**Figure 3 micromachines-15-01171-f003:**
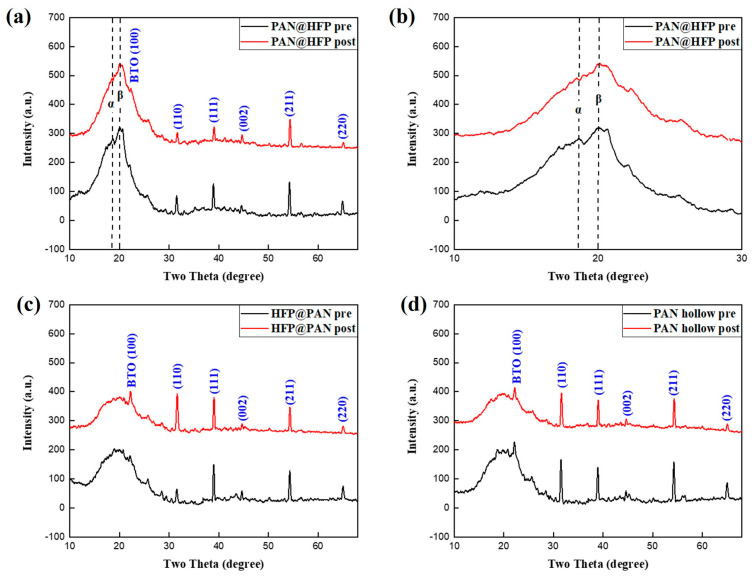
XRD images before and after cyclic test of (**a**) PAN@HFP, (**b**) partial enlargement, (**c**) HFP@PAN, and (**d**) PAN hollow.

**Figure 4 micromachines-15-01171-f004:**
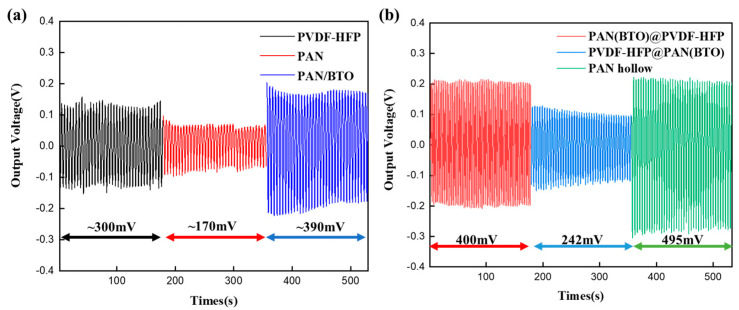
Output voltage (piezoelectric-induced) of (**a**) electrospun PVDF-HFP, PAN, BTO-doped PAN single nanofiber and (**b**) the coaxially structured fibers.

**Figure 5 micromachines-15-01171-f005:**
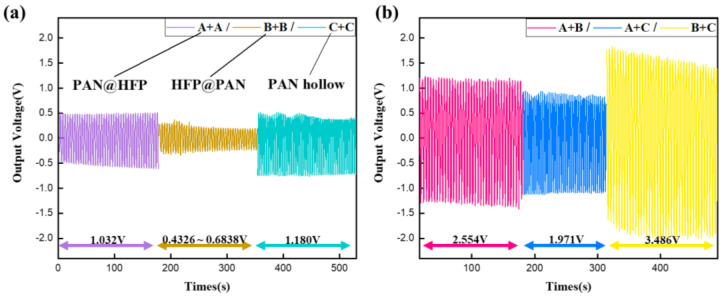
Output voltage (piezoelectric- and triboelectric-induced) of the hybrid nanogenerator; (**a**) homo group (measuring with two meshes of the same fibers), (**b**) hetero group (measuring with two meshes of the different fibers).

**Figure 6 micromachines-15-01171-f006:**
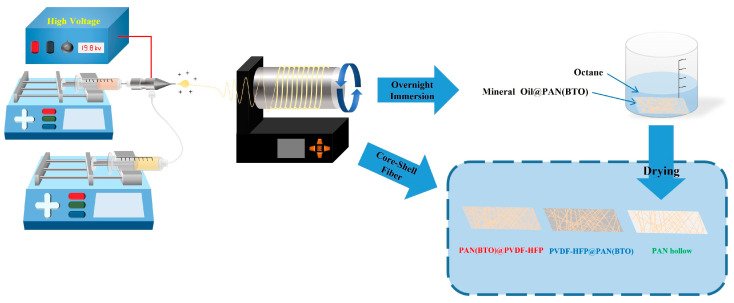
Schematic diagram of coaxial fiber electrospinning fabrication.

**Figure 7 micromachines-15-01171-f007:**
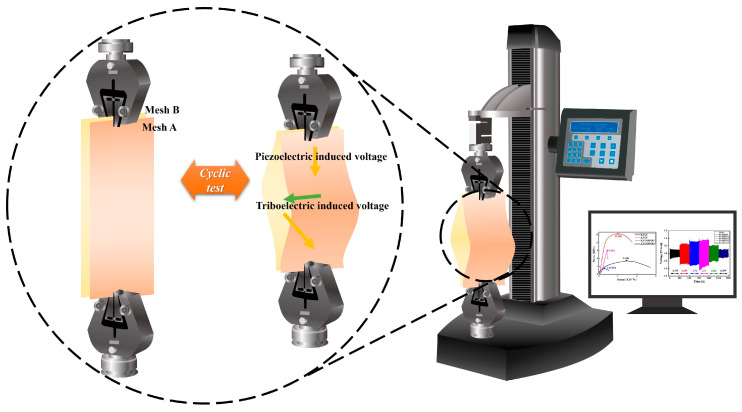
Piezoelectric- and triboelectric-induced voltage by tensile strength.

**Table 1 micromachines-15-01171-t001:** Hybrid nanogenerator voltage feature measurement.

Combination	A + A	B + B	C + C	A + B	A + C	B + C
Range of Output Voltage	0.958~1.107 V	0.433~0.612 V	1.098~1.262 V	2.513~2.596 V	1.883~2.059 V	3.431~3.541 V
Median of Voltage	1.033 V	0.523 V	1.180 V	2.555 V	1.971 V	3.486 V
Magnification	1.198~1.384	0.894~1.264	1.109~1.275	3.914~4.044	2.104~2.301	4.655~4.805

## Data Availability

The data presented in this study are available on request from the corresponding author.

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
