# Peer review of "Enhanced Hybrid Nanogenerator Based on PVDF-HFP and PAN/BTO Coaxially Structured Electrospun Nanofiber"

_micromachines, 2024, doi:10.3390/mi15091171_

Round 1
Reviewer 1 Report
Comments and Suggestions for Authors
This manuscript investigates hybrid nanogenerators based on coaxially structured electrospun nanofibers, focusing on the piezoelectric and triboelectric effects for energy harvesting applications. The study offers interesting advancements in material combinations, particularly with PVDF-HFP@PAN (BTO), presenting strategies in improving output voltage. However, several issues need to be addressed before considering acceptance:
- The dispersion of Barium Titanate (BTO) is eesential for enhancing the piezoelectric effect. While the manuscript describes successful BTO doping, the SEM image in Figure 1b does not clearly demonstrate whether the BTO is well dispersed throughout the fiber. In particular, the granular surface observed might suggest aggregation, which could undermine the uniformity of the piezoelectric response. The authors should provide a more detailed analysis of BTO distribution and possibly quantify the degree of dispersion. Uniform dispersion is key to maximizing the performance of nanogenerators.
- The manuscript mentions that the triboelectric effect between different polymers was more effective in generating voltage. However, a more detailed explanation of the fundamental mechanisms behind this observation would be beneficial to readers who are less familiar with triboelectric phenomena.
- In Figure 4, the output voltage of PVDF-HFP@PAN(BTO) decreases significantly over time, raising concerns about the long-term stability of this hybrid structure. The authors need to provide an explanation for this behavior, possibly addressing degradation mechanisms or the mechanical stability of the fibers. A more thorough investigation into the cause of this decline is needed, as it impacts the reliability and potential applications of the nanogenerator.
- The study discusses the successful conversion of mechanical energy into electrical energy using piezoelectric and triboelectric effects, yet no clear application is proposed. It would strengthen the manuscript to demonstrate a practical application, such as powering an LED or charging a capacitor. This would provide a concrete demonstration of the energy harvesting potential and connect the experimental results with real-world usage. Such an addition would significantly enhance the impact and relevance of the work.
- Please review the formatting of the references to ensure they adhere to the journal’s guidelines. Consistency in citation style is important for clarity and professionalism in the manuscript.
- Some of the SEM images in the figures could be more clearly labeled and referenced within the text. It may help to provide additional magnifications or contrasts that could reveal structural details more effectively.
Addressing these concerns will improve the manuscript’s overall clarity and reliability, potentially making it more impactful in the field of nanogenerators and energy harvesting.
Author Response
We appreciate the referee’s comments and suggestions. After reading the comments from Reviewers carefully, all the authors feel that the questions raised can be properly answered with acceptable revision. We, therefore, submit the revised manuscript for publication in Micromachines. All the changes made in revised manuscript are in red fonts.
Please Check our response letter

Reviewer 2 Report
Comments and Suggestions for Authors
- Title and Abstract Clarity:
- The title “Enhanced Hybrid Nanogenerator Based on PVDF-HFP and PAN/BTO Coaxially Structured Electrospun Nanofiber” is descriptive but quite long. Consider shortening it while maintaining specificity, perhaps by focusing on the key innovation or finding, like “High-Performance Hybrid Nanogenerator with Coaxial Electrospun Fibers.”
- Your abstract concisely summarizes the research’s goal and findings. However, adding a sentence about the practical implications or potential applications of these hybrid nanogenerators could provide more context and draw in a wider readership.
- Introduction:
- The introduction effectively sets up the background and importance of the research. It would be beneficial to explicitly state the hypothesis or primary research question earlier to guide the reader through the subsequent sections.
- Experimental Section:
- The description of the materials and methods is detailed, which is excellent for reproducibility. However, consider using subheadings to separate different parts of the experimental process (e.g., “Material Preparation,” “Electrospinning Process,” “Characterization Techniques”) for better readability.
- Results and Discussion:
- This section is well-articulated with supportive images and data. Enhancing the discussion on how your findings compare with existing technologies or previous studies could strengthen the manuscript’s impact. Cite recent studies for comparison where you discuss fiber diameter uniformity, voltage output, and material properties.
- The discussion on the implications of your findings on energy harvesting technology is good but could be expanded to include potential scale-up or real-world applications.
- Figures and Tables:
- Ensure that all figures and tables are clearly labeled and referenced in the text. Figure and table captions should be informative enough that the reader can understand the figure/table without extensive reference to the text.
- Consider adding a schematic diagram of the nanogenerator’s setup or operation if not already included, as this can help readers visualize the system’s function.
- Conclusion:
- The conclusions succinctly summarize the findings. You might also want to add recommendations for future research or mention any ongoing work to address current limitations.
- References:
- Check that all references are up-to-date and relevant, and ensure proper formatting as per journal guidelines. Consider including more recent references to demonstrate that you are fully engaged with current research.
- Language and Style:
- The document is well-written but review it for minor grammatical errors and stylistic consistency. Use of active voice where possible can make the text more engaging.
Here I have some questions:
(1) What are the key factors influencing the output voltage characteristics of hybrid nanogenerators using coaxially structured electrospun nanofibers, specifically in the context of material combinations like PVDF-HFP and PAN?
(2)How does the difference in the full width at half maximum (FWHM) of diffraction peaks between PVDF-HFP and 𝐵𝑎𝑇𝑖𝑂₃ impact the overall piezoelectric and triboelectric performance of the fibers?
(3) Can you elaborate on the role of the coaxial nozzle tip distance and voltage settings in optimizing the structural and electrical properties of PANBTO hollow fibers during electrospinning?
(4) What were the specific mechanical characteristics observed in the cyclic tensile tests that led to the differences in the output voltage of PVDF-HFP@PAN(BTO) compared to other hybrid combinations?
(5) Discuss the impact of environmental factors such as humidity and temperature on the electrospinning process and the resulting fiber properties in the context of the experimental setup described in the document.
(6) Given the study’s results, how could modifying the molecular weight distribution of the PVDF-HFP influence the efficiency and durability of the nanofibers?
(7) What challenges arise in scaling the production of these nanogenerators from a laboratory setting to industrial production, particularly in terms of maintaining uniformity and quality control of the fiber diameters and compositions?
(8) This report lacks scientific issues. Like an experimental report, can the content be supplemented with scientific significance?
Overall, your manuscript is robust and presents important findings in the field of materials science for energy harvesting. I suggest accept directly after minor revisions.
Comments on the Quality of English LanguageModerate editing of English language required.
Author Response

(The authors gave the same response as above.)

Round 2
Reviewer 1 Report
Comments and Suggestions for Authors
I have reviewed the revised manuscript, and I am pleased to report that the authors have addressed the comments I previously raised in the first revision. They have made the necessary updates, and the manuscript has improved significantly. I am satisfied with the revisions and believe that the authors have responded adequately to all concerns.
I recommend the manuscript for further consideration.